# Research on the competitive situation and technical opportunities of global new textile materials technology R&D

**Pengfei Wang** [1,2]*, **Hua Cheng**[1,2]

**1** School of Economics and Management, Zhejiang Sci-Tech University, Hangzhou, China, **2** Key Laboratory of Intelligent Textile and Flexible Interconnection of Zhejiang Province, Hangzhou, China

* wangpf@zstu.edu.cn

## Abstract

Vigorously developing new textile materials technology is of great significance to improve the high-tech level of the textile industry, enhance the transformation and upgrading of the textile industry, promoting the sustainable development of the industry and the comprehensive national strength. This paper takes the 94,525 invention patents for new textile material technology in the world as the research object, and uses competitive situation calculation and patentometrics to explore the competitive situation, technical opportunities, and layout characteristics of technology R&D (Research and Development) in this field. The results show that the core patented technologies of the United States in the field of new textile materials are in a global leading position with a strong sense of overseas layout, which, like Japan, is more forward-looking than other countries and regions. The latter occupies a pivotal international position in a number of cutting-edge materials such as fibers, functional chemicals, and carbon fiber composites. Although China is a global patent major country, there is still a huge gap between the core patent R&D capabilities and the awareness of technical opportunities compared with developed countries such as the United States and Japan, which is the main problem that needs to be solved urgently in the future innovation and development, technology planning and layout of China's textile industry. This study effectively supplement the theoretical analysis of competitive situation and technical opportunities of technology R&D, and enrich and expand the empirical research on international new textile materials technology and innovation.

## 1. Introduction

Materials are the basic elements of all social life and economic development of mankind, and as a key resource input, they play a vital role in promoting the progress of the technological revolution [1]. New materials refer to new materials with excellent properties or special functions, or materials with significantly improved performance or new functions after the improvement of traditional materials. As a basic and pillar strategic industry, new materials are the forerunner and cornerstone of high-tech development, also the indispensable material foundation for the development of the national economy, national defense, and other high-tech industries, playing an important role in promoting technological innovation, accelerating

**Data availability statement:** The data used in this paper was obtained from the PatSnap database (https://analytics.zhihuiya.com/).

**Funding:** This study was supported by the Project Fund of Key Laboratory of Intelligent Textile and Flexible Interconnection of Zhejiang Province (Project No.111504A4E22004), and Silk & Fashion Culture Research Center of Zhejiang Province, Zhejiang Provincial Key Research Institute of Philosophy and Social Science (Project No. 2022JDKTYB27). The funders had no role in study design, data collection and analysis, decision to publish, or preparation of the manuscript.

**Competing interests:** The authors have declared that no competing interests exist.

the optimization and upgrading the manufacturing industry, and ensuring national security [2,3]. In recent years, with the rapid development of human society and science and technology, more and more countries have taken the development of the new textile material industry as a major national strategic decision. Such a new textile technology of multidisciplinary convergence and integration has become the main direction to drive the upgrading of the textile industry chain, enhance the competitive advantage of enterprises, and guide the innovation and development of textile science and technology.

With the progress of society and the development of science and technology, many industries have applied new textile materials, which mainly refer to the use of advanced technology and equipment to produce high-tech fibers, high-tech fabrics and high-tech industrial textiles. The R&D of new textile materials is increasingly important to maintain the sustainable development of society as green, environmental protection, energy conservation, and emission reduction have become the common goals of the world. As a key area of international competition, new textile materials technology is a key force in promoting the high-end and intelligent development of the textile industry, and it is also an important engine for the realization of a modern textile industrial system. Therefore, an in-depth study of the global textile new materials technology competition situation, analysis of the development status of key countries, and further recognition of the technical layout and potential technical opportunities in this field will help China to carry out forward-looking and strategic technology research and development and layout with positive guiding significance for promoting technological innovation and sustainable development of the textile industry.

Patent measurement is a method to comprehensively understand the level, the new emerging clustering technique and trend of industrial technology development, and technology measurement analysis with patentometrics as the main method is one of the effective and practical methods, including examining the innovation activities of firms in the textile and clothing industry [4–6].

However, there is a lack of research of the competitive situation and technical opportunities of global new textile materials technology R&D, and the empirical study on international new textile materials technology and innovation can provide a more accurate and comprehensive panoramic analysis, serving a reference for the government's science and technology innovation policy formulation, technology layout and R&D. Therefore, this paper attempts to supplement the existing research in terms of research object and research content, to potentially fill the gap of existing research.

The significance of this study is as follows: (1) Laying a theoretical foundation for future emerging technology cooperation in R&D and innovation. (2) Emphasize the findings can inform industry practices, R&D strategies, and policy frameworks. (3) Through the empirical analysis of global patent data, it provides a reference and basis for the government to rationally allocate innovation resources and formulate relevant policies, which has practical significance and application value for the decision-making of enterprise technology R&D departments.

The possible marginal contributions of this paper are as follows: (1) Supplementation of the research framework. Based on the 94,525 invention patents for new textile material technology in the world as the research object, this paper constructs the the competitive situation, technical opportunities, and layout characteristics of 170 countries and regions around the world patent technology innovation in order to effectively supplement the analytical framework of international technology R&D. (2) Expansion of research methods. This paper uses competitive situation calculation and patentometrics to scientifically explore the characteristic factors of technology R&D ability and technical opportunities so as to enrich and expand the

empirical research on international technology innovation and provide policy enlightenment for optimizing regional sustainable development and national R&D innovation.

## 2. Literature Review

At present, scholars' research on the technological innovation of new textile materials mainly focuses on three aspects: (1) In the field of textile raw material technology, such as herbal biomass, natural dyes, and extracted cellulose and fiber extraction [7], which are widely used in the development of high-performance, smart textiles, and multi-functional materials field [8,9], which is to meet the needs of the industry and the market, and improve the added value and competitiveness of textiles. (2) Case studies focusing on technological innovations in new textile materials, such as the use of eco-friendly technologies to produce textiles with the required physicochemical, woven auxetic fabrics and mechanical properties field [10–12], as well as scalable nanofabrication technologies to tailor smart fabrics [13] and natural fabrics [14], to meet various potential application needs. (3) Analysis of the policy mechanism of technological innovation of new textile materials, evaluates the expected impact of new technologies on topics such as the environment, automotive, aerospace, and ballistic areas and human security [15–17], and emphasizes the important role of policy guidance and supervision in the sustainable development of technology. In addition, artificial intelligence (AI) technology is also gaining more and more attention in the apparel industries, and researchers are constantly exploring and optimizing the use of AI techniques in a range of industrial applications [18,19]. Since patents are synonymous with innovative activities, the analysis of new textile material technology through patents has also attracted the attention of many scholars [20].

Core patents reflect the value of patented technologies, and the corresponding technologies or products can occupy market share and obtain high economic benefits by their competitiveness, so they have a high degree of technological competitive advantage and market economic value [21,22]. At present, academic research focuses more on the technical perspective, pointing out that the core patents are the technical patents that must be used to manufacture a certain technical field or a certain product without circumvention. Some scholars also believe that core patents are patents or patent portfolios with an important economic and strategic value that are in a key position of innovation, influential, and inevitable in a certain technical field [23–25]. Based on this research, this paper defines core patents as patents that are in a fundamental position in the technological evolution route, absolutely leading and contributing to the subsequent development of the technology with a high reference value for epitaxial technology.

The research results on technological opportunities were first proposed by Porter et al. [26] in 1995, and its core is to conduct in-depth text mining of academic papers, patent information and network resources, etc., to identify the interrelationships and development trends of existing technologies, to grasp the latest technological trends, and provide support for future technological innovation of enterprises, which also shows to a certain extent that technology opportunity analysis is essentially the mining of technological knowledge. At present, many scholars have conducted and achieved a series of results on the research in the field of artificial intelligence at different levels from the perspective of scientific literature, involving the knowledge structure, research strength, and future technological trends of expert systems [27–29]. As a special scientific document, patents reflect the development of technological innovation as an important output of scientific research and innovation activities [30]. Through the mining of patent literature, we can better identify and grasp the potential opportunities in future technology development. In this regard, this paper identifies and analyzes the technical

opportunities in the field of new textile materials from three dimensions: innovation subject, patent layout, and technology integration, to provide a useful reference for the government, industry, and research institutions.

In summary, the current research on new textile material technology is mostly aimed at the new textile industry, AI technologies in the apparel industry [31] and the technical field at the level of new textiles, and there are few studies on the competitive situation and technical opportunity analysis of international textile new material technology R&D in academic circles, especially the lack of combining global patent data comparative research. Therefore, this paper takes 94,525 invention patents of new textile material technology in the world as the research object and explores the competition situation, layout characteristics, development trends, and technical opportunities of new textile material technology in major countries and regions in the world, to provide a useful reference for the technological innovation and development of the textile industry.

## 3. Theoretical analysis

Patent data mining is the use of patent analysis to extract valuable data information from patent documents, such as the state of industrial technology development, the main areas of technological competition and competitors' technology layout. Through patent competitive intelligence, it helps enterprises understand the industry environment, evaluate themselves and monitor their competitors so as to obtain competitive advantages and potential opportunities [32–34].

In recent years, technology competition in emerging industries has become increasingly fierce, and more and more scholars have identified the main competitors in specific technology fields from the perspective of patent competition analysis or patent cooperation analysis, and helped innovation subjects to construct patent strategies and plans to achieve the goal of enhancing competitive advantage by tracking the trend of technology development and evaluating the innovation ability of technology [35]. With the rapid development of digital technology, the innovation ability of enterprises has become an important measure of national competitiveness. Therefore, the policy influence of different countries and the research suggestions of scholars have also had a positive effect on the decisive factors for innovative subjects to gain competitive advantages in the market, and provide an important reference for the country to formulate relevant technological strategies and promote the transformation of scientific and technological achievements [36].

At the same time, as the premise of technology R&D and innovation, technology opportunity research has become a key issue of concern to the state and enterprises, and has important strategic significance for technology competition, market layout, regional innovation and industrial development. Therefore, mining technological opportunities from patent data can not only obtain possible new technologies, or new technologies that are improved on the basis of existing technologies, but also provide enterprises with important R&D intelligence and reference directions through analysis [37–39].

To sum up, the continuous innovation of technology will promote the development of the industry, this process on the one hand requires the progress of technology itself, on the other hand, there is also a demand from the market, the two need to reach a technical research and development in the unity of technical and economic possibilities [40]. Therefore, on the basis of existing research, this paper analyzes the competitive situation and technical opportunities of global textile new material technology research and development in the 21st century, in order to provide a useful reference for the forward-looking and strategic research and development layout of the government, industry and enterprise technology innovation.

## 4. Methods

### 4.1. Competitive situation calculation process

With the implementation of the economic globalization strategy and the rapid development of textile technology, a large number of multinational enterprises participate in the international market competition, and enterprises rely on applying for patents in different countries and regions to improve the competitiveness of products and technologies. Therefore, the use of core patents to carry out overseas patent layouts can help enterprises obtain exclusive intellectual property rights, and play a positive role in promoting enterprises to build brand value and enhance technological innovation capabilities [41]. At the same time, based on the recognition and respect for intellectual property rights, products with overseas intellectual property protection may gain more recognition in the process of expanding overseas markets. To further analyze the layout of each country in the global market of new textile material technology, this paper determines the core patents according to Price law and calculates the overseas export rate (OER) and the acceptance layout rate (ALR) of a country or region according to the formula in Table 1.

### 4.2. Patentometrics

Information such as the number of patents, patent types, patentees, and patent composition categories reflect the development trend of technological innovation in the industry. The general trend of the development of the technology field can be obtained by using the method of patent measurement to analyze the technical characteristics of the sample data [42–44]. This paper uses the global textile new material technology patent application data to reveal the characteristics and competitive layout of global textile new material technology innovation from the perspective of technology research and development competition situation and technology opportunity analysis and provides reference and selection basis for the innovation and development of the textile industry.

### 4.3. Data sources

The data used in this paper was obtained from the PatSnap database (https://analytics.zhihuiya.com/). For the new textile material technology, according to the existing research results [45–47], it is mainly defined as high-tech fibers, high-tech fabrics, and high-tech industrial textiles produced by high-tech processes and equipment manufacturing in recent years. Its International Patent Classification (IPC) includes two categories: D (textile) and A41 (apparel). Therefore, the new material industry is selected in the national strategic emerging industry classification, and combined with the IPC D01 fiber, D02 yarn, D03 weaving, D04

**Table 1. Calculation formula and description of technology R&D competitive situation indicators.**

| Patent indicators | formula | description |
|---|---|---|
| Core patents(C) | $\mathrm{INT}\left(0.749 \times \sqrt{N_{max}}\right)$ | $N_{max}$: The maximum number of citations for all patents; INT (): The value of citation frequency greater than or equal to the result C is determined as the core patent after taking the integer calculation. |
| Patent layout | $\mathrm{OER_i} = \dfrac{\sum P_{(i \to j)}}{P_i} \times \%$ | $P_{(i \to j)}$: Number of patent applications for new textile material technologies filed by country or region i in country or region j. |
| | $\mathrm{ALR_i} = \dfrac{\sum P_{(j \to i)}}{P_i} \times \%$ | $P_{(j \to i)}$: Number of patent applications for new textile material technologies received by country or region i from country or region j. |

knitting, D05 sewing, and D06 fabric treatment, etc., the patent legal status is selected as "valid". Since the period from invention patent application to publication and authorization is about 18 months, the search time frame is limited to 2021, and 170 countries and regions around the world were selected as the innovation areas of research. Through the above settings, the study was carried out, and then the results were combined with the applications, and 94,525 patent data consistent with this study were finally screened.

## 5. Empirical analysis results

### 5.1. Analysis of the competitive situation of new textile material technology R&D

**Patentee analysis.** The main patentees in the sample data were selected for analysis, and the distribution data table of the top 14 patentees shown in Table 2 was obtained. Among them, the performance of Japanese companies is particularly outstanding, including Toray Corporation, Kao Corporation, Kuraray Co., Ltd., and Mitsubishi Chemical Corporation, which are seven internationally renowned companies and multinational enterprises. As a world-renowned high-tech enterprise with organic synthesis, polymer chemistry, and biochemistry as its core technologies, Toray Co., Ltd., ranks first with 2,037 patents in new textile materials technology, which is almost next followed by Procter & Gamble(1,281), and Donghua University(920) combined. The company has a large number of patent applications in China, the United States, South Korea, and Europe, reflecting its international position in many cutting-edge materials such as fibers, functional chemicals, and carbon fiber composites. In addition, there are three cross-border enterprises in the United States, including Procter & Gamble, Kimberly Clark Global Co., Ltd., and Dow Global Technology Co., Ltd., and two in China and South Korea. Above three patentees constitute the main body of innovation in the world's new textile material technology.

An analysis of the filing time of the world's major patentees shows that the layout of new textile material technology in the United States and Japan is more forward-looking than that of other countries and regions. Kao Co., Ltd. and Kimberly-Clark Global Co., Ltd. began to apply as early as 2001, and Procter & Gamble and Kuraray Co., Ltd. also began to deploy in

**Table 2. Major global patentees of new textile material technologies.**

| No. | patentees | Number of patents | Country | Time of first application (number) |
|---|---|---|---|---|
| 1 | Toray Co., Ltd. | 2037 | Japan | 2003 (5) |
| 2 | Procter & Gamble | 1281 | United States | 2002 (2) |
| 3 | Donghua University | 920 | China | 2004 (1) |
| 4 | Kolon Industries, Ltd | 868 | South Korea | 2004 (18) |
| 5 | Kao Corporation | 858 | Japan | 2001 (1) |
| 6 | Kimberly Clark Global Co., Ltd. | 780 | United States | 2001 (1) |
| 7 | Kuraray Co., Ltd | 738 | Japan | 2002 (1) |
| 8 | Mitsubishi Chemical Corporation | 719 | Japan | 2004 (3) |
| 9 | Asahi Kasei Co., Ltd | 559 | Japan | 2005 (4) |
| 10 | Dow Global Technology Co., Ltd. | 532 | United States | 2003 (2) |
| 11 | Teijin Corporation | 490 | Japan | 2003 (1) |
| 12 | Mitsui Chemicals Corporation | 489 | Japan | 2004 (26) |
| 13 | Jiangsu Hengli Chemical Fiber Co., Ltd | 487 | China | 2008 (2) |
| 14 | LG Chem Corporation | 431 | South Korea | 2005 (1) |

2002. The application time of Chinese and Korean enterprises is relatively late, and Jiangsu Hengli Chemical Fiber Co., Ltd. did not have application data until 2008. As an important development force in the development of new textile material technology, with the support of government policies and the integration of resources, the two countries have developed rapidly, especially Kolon Industries, which applied for 18 patents in 2004, reflecting its strong innovation strength and competitiveness in this field.

**Analysis of core patents.** Core patents reflect the value of patented technology, which is an effective way to measure patent capital. The citation frequency reflects the degree to which the patent is used and valued, showing the influence of the patent [48,49]. Patent citation reflects the development law of the technical field, indicating the continuity and succession of science and technology, and the cross-integration between different disciplines. The number of patent families and the frequency of citations are commonly used indicators in patent econometrics [50]. Analyzing the citation frequency of patents to evaluate the citation of patents not only reflects the external performance of patents as expansive and groundbreaking but also illustrates the positive relationship between the higher the number of citations of patents and the greater the technical influence of the patent [51] from the perspective of technology accumulation and innovation. The patent system is territorial, and the more countries that apply for patents for the same technology, the wider the geographical scope and the higher the corresponding value [52]. Therefore, this paper synthesizes the number of patent citations, the number of patents in the same family, and the number of countries/regions in the layout to obtain the top 10 core patents of global textile new materials technology as shown in Table 3.

By analyzing the number of citations of patents from data sources, the highest is 795 times with a patent number US9011439B2, that is, $N_{max} = 795$, so with the help of the above core patent calculation formula, $C = 21$ is obtained, the total number of core patents cited more than or equal to 21 times is 712. Among the top 10 core patents in Table 3, the number of first applicants from the United States reached 6, of which Bambu Vault LLC held 2 core patents. The seven high-quality patents also reflect the global leadership of the United States in the field of new textile material technology, showing a state of "one of the best". The two first applicants selected by China are Donghua University and Tsinghua University, indicating

Table 3. Global top 10 core patents of new textile material technologies.

| No. | Patent Number | First applicant | Number of citations | Number of patents in the same family | Number of countries/regions in the layout |
|---|---|---|---|---|---|
| 1 | US9011439B2 | Poly-Med, Inc. | 795 | 3 | 1 |
| 2 | CN102783741B | Donghua University | 589 | 1 | 1 |
| 3 | US8207511B2 | Bambu Vault LLC | 491 | 7 | 4 |
| 4 | US7045108B2 | Hon Hai Precision Industry Co., Ltd. | 464 | 3 | 3 |
| 5 | US7172801B2 | Procter & Gamble Co. | 366 | 35 | 17 |
| 6 | US8846184B2 | Bambu Vault LLC | 315 | 4 | 1 |
| 7 | US8221669B2 | Stratasys, Inc. | 308 | 16 | 12 |
| 8 | CN100500556C | Tsinghua University | 304 | 5 | 3 |
| 9 | US7314497B2 | Donaldson Co., Inc | 292 | 56 | 17 |
| 10 | KR101032792B1 | Kolon GROUP | 265 | 1 | 1 |

that universities play an important role in the field of textile technology innovation, and an important part of the national industry-university-research collaborative innovation ecosystem [53], colleges and universities play a pivotal role in technology R&D and sustainable industrial development.

In addition, the United States far exceeds other countries in the number of patent families, represented by Procter & Gamble Co., Donaldson Co., Inc. and Stratasys, Inc., etc., with a strong sense of overseas layout, which truly reflects the characteristics of "big and strong, many and excellent" in the field of new textile material technology. Although it has been proposed that the number of patent citations is used as an index to judge the value of patents, which is difficult to objectively reflect the value of patents [54], it cannot be denied that the United States has a large number of core patents in new textile material technology, reflecting its global leadership and competitiveness in textile technology innovation and development.

**Technical field analysis.** In this paper, combined with the IPC (International Patent Classification), the first 4 digits of the code IPC4 are selected, and patent analysis of the global and Chinese focus areas from a technical perspective can be carried out, to be able to explore the current important technical fields of new textile materials from a micro perspective, and the results as shown in Table 4. China and other countries in the world are relatively more concerned about the technology that mainly includes D01F (equipment for making artificial filament fiber chemical characteristics or carbon fiber), D06M (biochemical treatment of fiber products made of fiber, yarn and thread raw materials), D04H (textiles manufacturing), and D01D (mechanical method or equipment for making chemical filament, thread and fiber), etc., illustrating the production of artificial filament, the chemical characteristics of fibers, equipment dedicated to the production of carbon fibers, manufacturing textiles and making chemical filaments, fibers, and other technologies are the most favored by the market. Moreover, the proportion of the above technologies in China is higher than the number in the world, indicating that the huge market demand and technology application have played a positive role in promoting the research development, and innovation of domestic textile technology, and have become a global research hotspot in the field of new textile materials.

**Table 4. Global distribution of new textile material technologies.**

| No. | Global | | | China | | |
|---|---|---|---|---|---|---|
| | IPC4 | Number of patents | rate (%) | IPC4 | Number of patents | rate (%) |
| 1 | D01F | 14801 | 24.63 | D01F | 10796 | 31.29 |
| 2 | D04H | 13686 | 22.78 | D06M | 9847 | 28.54 |
| 3 | D06M | 11961 | 19.90 | D01D | 6979 | 20.23 |
| 4 | D01D | 7984 | 13.29 | D04H | 4127 | 11.96 |
| 5 | D03D | 5927 | 9.86 | D06P | 3945 | 11.44 |
| 6 | C08L | 5110 | 8.50 | C08G | 3408 | 9.88 |
| 7 | D02G | 5040 | 8.39 | D02G | 2768 | 8.02 |
| 8 | C08J | 4498 | 7.49 | D03D | 2309 | 6.69 |
| 9 | C08K | 3245 | 5.40 | C08L | 2296 | 6.66 |
| 10 | C08G | 3137 | 5.22 | D06B | 1647 | 4.77 |
| 11 | D06P | 3088 | 5.14 | C08K | 1595 | 4.62 |
| 12 | C09D | 2997 | 4.99 | C08F | 1503 | 4.36 |
| 13 | A41D | 2961 | 4.93 | C08J | 1411 | 4.09 |
| 14 | B01D | 2320 | 3.86 | D06C | 1410 | 4.09 |
| 15 | C08F | 1859 | 3.09 | C09B | 1161 | 3.37 |
| 16 | D04B | 1762 | 2.93 | D04B | 1139 | 3.30 |

Although there are similarities in the distribution of new textile material technologies at home and abroad, obvious differences exist. C08L (polymer compound compositions), A41D (outerwear, protective clothing, apparel accessories), and C09D (paint compositions, e.g., colour paints, varnishes or natural paints) technologies also occupy a relatively important share in the world, indicating that the research on the composition of polymer compounds, outerwear, and coating compositions has also received great attention from foreign enterprises. D06C (fabric finishing, sizing, stretching or elongation), C09B (organic dyes), and D04B (knitted) technologies also have a high ratio in China, reflecting the strong demand in the domestic market for fabric finishing, organic dyes, or related compounds used to make dyes and knitting technologies.

## 5.2. Analysis of new textile materials technology opportunities

**Analysis of technology opportunities from the perspective of innovation subjects.** Discovering potential competitors and their advantageous directions from the perspective of the innovator can help enterprises formulate technological innovation strategies and discover or expand technological opportunities. Therefore, this paper analyzes the distribution of the strength of innovative subjects in the field of textile new material technology from the macro level of the country (region) and the micro level of technology that enterprises are concerned about. The results as shown in Table 5. At present, the patents of new textile material technology are mainly distributed in East Asia and North America. In terms of quantity, China ranks first with 34,481 patents, Japan (13,347), the United States (11,467), and South Korea (10,438) are in the same echelon, but the development of this field is very uneven in the world, and the number of patents applied for by China, the United States and Japan and South Korea exceeds 70% of the global total, showing the obvious strength and advantages of technological innovation.

The results of the World Intellectual Property Organization's (WIPO) Global Innovation Index (GII) 2022 showed that the world's top five science and technology clusters are composed of China, Japan, South Korea, and the United States, which is also a strong validation of the above conclusions of innovation subjects. As the leading force in scientific and technological innovation, the competition for national scientific and technological strength is reflected in the competition between enterprises at the micro level, so the technology-patentee R&D network reflects the patent layout of global research institutions. The results as shown in Fig 1. Among them, the polygon icon represents the IPC4 patent classification number of textile new material technology, and the square icon represents the R&D institution.

Table 5. Global distribution of new textile material technologies by country and region.

| No. | Country (Region) | Number of patents | rate (%) | Cumulative proportion (%) |
|---|---|---|---|---|
| 1 | China | 34481 | 36.48 | 36.48 |
| 2 | Japan | 13347 | 14.12 | 50.60 |
| 3 | United States | 11467 | 12.13 | 62.73 |
| 4 | South Korea | 10438 | 11.04 | 73.77 |
| 5 | European Patent Office | 6877 | 7.28 | 81.05 |
| 6 | Chinese Taiwan | 2502 | 2.65 | 87.70 |
| 7 | Spain | 1703 | 1.80 | 85.50 |
| 8 | Canda | 1473 | 1.56 | 87.06 |
| 9 | Brazil | 1174 | 1.24 | 88.30 |
| 10 | Others | 11063 | 11.70 | 100.00 |

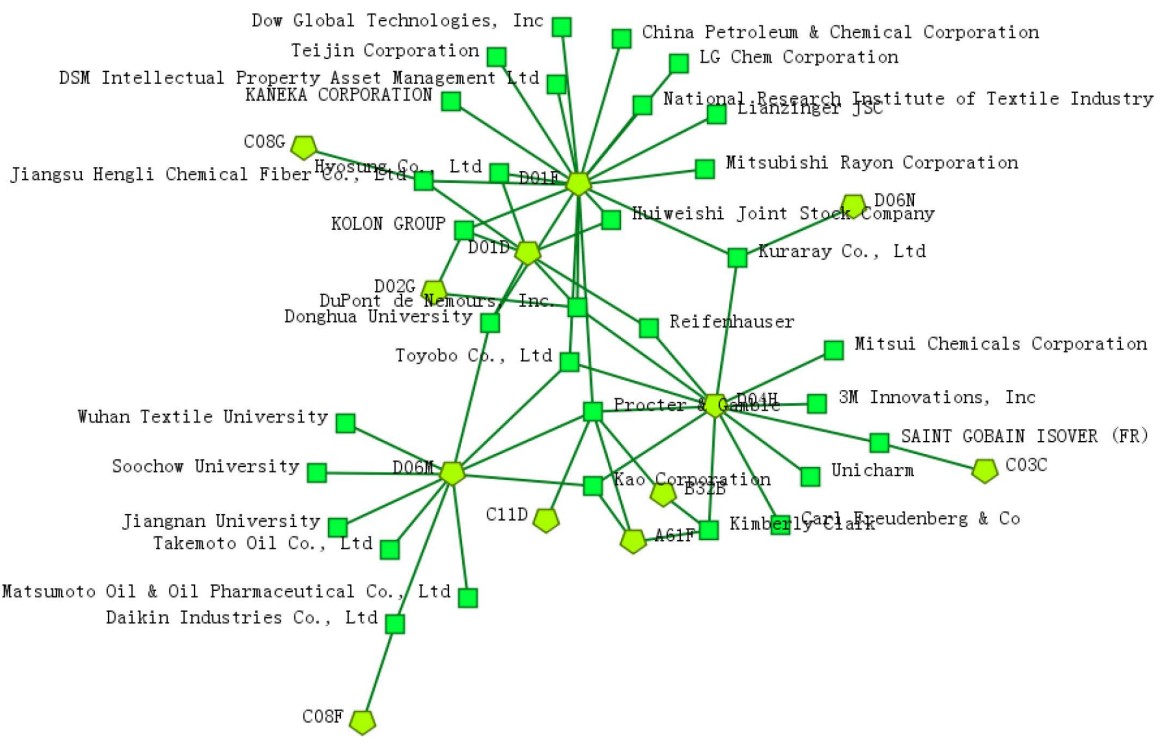

**Fig 1. "Technology-patentee" R&D network.**

Fig 1 shows that technologies such as D01F, D06M, D04H, and D01D are at the core of the R&D network and have become the main direction of attention and layout of many R&D institutions. Among them, Kuraray Co., Ltd mainly focuses on D01F and D04H technologies, and as a well-known comprehensive chemical company in Japan, it has established a high-quality technology system with originality based on polymer chemistry and synthetic chemistry technology, integrating related peripheral technologies and experience. In addition, multinational companies such as Procter & Gamble, DuPont, and Kolon Industries, Ltd also have a wide range of layouts in D02G (fiber) technology.

**Analysis of technology opportunities from the perspective of patent portfolios.** To protect the main market or potential market of existing technologies and products, innovative entities carry out overseas patent layouts by applying for patents. Therefore, the analysis of the overseas layout of patents in different countries or regions is helpful in digging out the layout characteristics of new textile material technology. Table 6 shows the matrix of the global patent layout of new textile material technology, indicating the number of patents developed in China (CN), the United States (US), Japan (JP), South Korea (KR), Canada (CA), Spain (ES), the United Kingdom (GB), South Africa (ZA), France (FR), Germany (DE) and Brazil (BR), and the last column shows the patent family ratio of the country's overseas layout.

Although China's textile new material technology patent applications rank first in the world, only 7.02% of the patents are applied for overseas at the same time, and the acceptance rate of the layout rate is 16.98%, of which the main source countries are Japan (40.39%), the United States (24.27%), Germany (9.68%) and South Korea (7.04%), etc. These countries apply for a relatively high share of patents in China, not only having a far-reaching impact on domestic textile technology innovation but also acting as an important importer of key core technologies in China's textile industry green production, smart manufacturing, and

**Table 6. Patent layout matrix of global textile new material technologies.**

| Country | CN | US | JP | KR | CA | ES | GB | ZA | FR | DE | … | BR | Sum | Oveasea proportion |
|---|---|---|---|---|---|---|---|---|---|---|---|---|---|---|
| CN | 32002 | 669 | 282 | 105 | 28 | 31 | 7 | 31 | 1 | 29 | … | 28 | 34496 | 7.02% |
| US | 1425 | 5455 | 1484 | 793 | 794 | 341 | 91 | 147 | 11 | 199 | … | 473 | 11467 | 86.89% |
| JP | 2377 | 2077 | 9812 | 1260 | 142 | 163 | 4 | 32 | 2 | 143 | … | 96 | 13339 | 74.15% |
| KR | 413 | 585 | 409 | 7425 | 17 | 26 | 6 | 9 | 6 | 43 | … | 23 | 10452 | 19.34% |
| CA | 16 | 117 | 11 | 8 | 54 | 13 | 1 | 3 | 1 | 1 | … | 1 | 311 | 82.64% |
| ES | 8 | 34 | 12 | 4 | 2 | 104 | 1 | 1 | 4 | 4 | … | 3 | 251 | 58.57% |
| GB | 66 | 179 | 77 | 42 | 28 | 54 | 86 | 27 | 3 | 12 | … | 37 | 978 | 91.21% |
| ZA | 1 | 4 | 1 | 0 | 0 | 0 | 0 | 8 | 0 | 0 | … | 0 | 22 | 63.64% |
| FR | 216 | 401 | 198 | 76 | 116 | 123 | 1 | 16 | 519 | 51 | … | 98 | 2754 | 81.15% |
| DE | 567 | 793 | 446 | 244 | 99 | 351 | 8 | 38 | 12 | 392 | … | 126 | 5430 | 91.22% |
| BR | 6 | 19 | 5 | 5 | 7 | 10 | 1 | 2 | 2 | 1 | … | 49 | 141 | 68.79% |

high-performance fibers. This not only shows that the Chinese market has a huge external attraction, but also shows that developed countries or regions such as Europe, the United States, and Japan use technological opportunities to carry out patent layouts and advanced strategies, forming a relatively strong control over the Chinese textile market. In contrast, China's innovation institutions are seriously lagging in their awareness and action of international patent layout, and are in a weak position in seizing the international market.

According to the global textile new material technology patent layout matrix, and with the help of the scientific knowledge graph software Gephi, the patent overseas layout network as shown in Fig 2 can be obtained. Among them, the United States (US), Japan (JP), China (CN), South Korea (KR), and Germany (DE) are located in the center of the network. Therefore, these countries are the key areas of the patent layout of new textile material technology, and they are also important markets that are closely watched.

**Analysis of technology opportunities from the perspective of technology convergence.** Modern science and technology present a dual development trend of high integration and high differentiation, while the correlation between different disciplines or technical fields is precisely the basis for the convergence and differentiation of science [55,56]. New textile materials are a highly comprehensive interdisciplinary discipline developed based on multidisciplinarity, involving materials science, chemistry, engineering, physics, biology, and computer science. Due to its convergence of multiple technologies, smart manufacturing, digital technology, chemical industry, biomanufacturing, and advanced material technology offer great opportunities for the development of new textile materials. As the most effective carrier of technical information, patents contain the latest and most comprehensive technical information in the world and are the most valuable indicators to measure technological development and innovation. Therefore, it is scientific and feasible to discuss the level of technology convergence based on patents.

Due to the cross-convergence of modern technology, it usually involves multiple technical fields. This study combines two aspects of breadth and depth to further analyze the degree of convergence between new textile material technologies around the world (the results are shown in Table 7. Among them, the breadth of technology convergence refers to the proportion of the number of technologies integrated with other technologies in the total number of technology topics, the greater the breadth, the more basic the technology; the depth of technology convergence refers to the proportion of the number of patents with technology integration under a certain technology theme to the number of all patents under the theme, and the higher the depth, the stronger the dependence of the technology on other technologies.

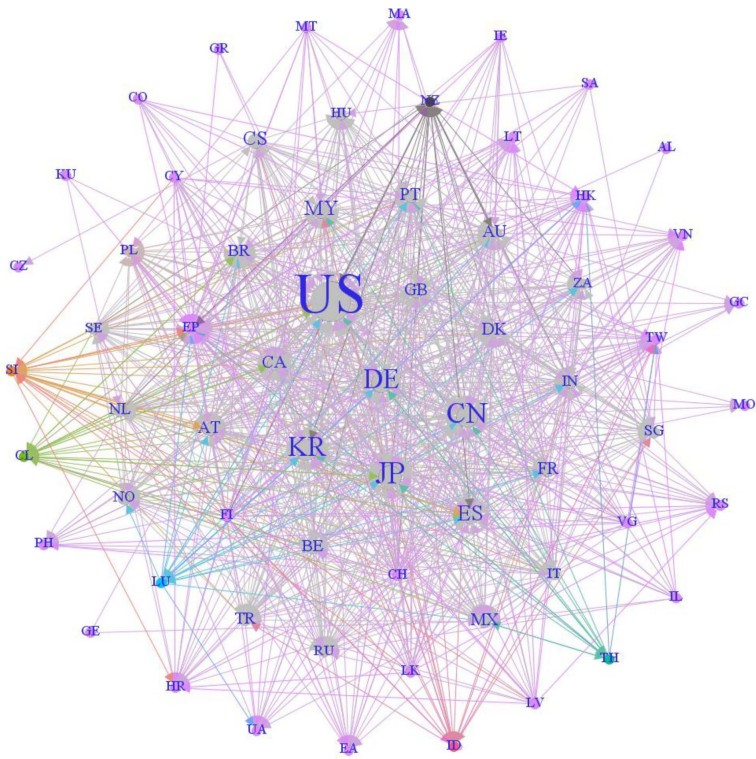

**Fig 2. Patent layout network of global textile new material technologies.**

**Table 7. The breadth and depth of global textile new material technology convergence.**

| IPC4 | global | | | China | | |
|---|---|---|---|---|---|---|
| | Number of patents | breadth of convergence (%) | depth of convergence (%) | Number of patents | breadth of convergence (%) | depth of convergence (%) |
| D01F | 14799 | 57.00 | 38.33 | 10791 | 42.08 | 43.37 |
| D06M | 11956 | 66.90 | 60.13 | 9841 | 55.10 | 66.94 |
| D01D | 7983 | 55.83 | 68.08 | 6979 | 42.24 | 66.96 |
| D04H | 13680 | 62.86 | 55.47 | 4124 | 41.61 | 52.66 |
| D06P | 3086 | 51.64 | 44.63 | 3942 | 56.39 | 65.21 |
| C08G | 3136 | 73.49 | 48.04 | 3406 | 46.40 | 67.60 |
| D02G | 5041 | 54.95 | 56.07 | 2767 | 40.04 | 60.20 |
| D03D | 5925 | 58.40 | 61.59 | 2309 | 43.20 | 41.09 |
| C08L | 5107 | 77.38 | 51.12 | 2294 | 62.54 | 43.37 |
| D04B | 1761 | 59.67 | 46.38 | 1139 | 41.98 | 53.19 |
| C08K | 3243 | 79.25 | 53.49 | 1593 | 59.26 | 40.68 |
| C08F | 1858 | 79.54 | 39.79 | 1501 | 57.51 | 55.41 |
| C08J | 4498 | 74.19 | 64.44 | 1410 | 59.17 | 35.81 |
| C09D | 2996 | 51.35 | 47.03 | 1046 | 74.66 | 28.51 |
| A41D | 2961 | 79.30 | 49.61 | 952 | 61.06 | 34.08 |
| B01D | 2319 | 52.26 | 65.64 | 833 | 51.80 | 36.25 |

Globally, the range of technology convergency is more than 70% of technologies such as C08F (a polymer compound obtained by reacting only with carbon-carbon unsaturated bonds), A41D, C08K (using inorganic or non-polymeric organic compounds as ingredients), C08L and C08G (polymer compounds obtained by reactions other than carbon-carbon unsaturated bonds), That is to say, these technologies are highly complex and closely related to other technologies in an important node position in the global textile new material technology network. However, there are obvious differences in the breadth of China's technology integration, and there is only one technology for more than 70%, indicating that the integration of China's new textile material technology is still a long way. In terms of the depth of fusion, the patented technologies represented by foreign D01D, B01D (separation), and C08J (processing, general process of ingredients) are more obvious. In contrast, China's technology convergence is mainly reflected in the technical themes of C08G, D01D, D06M, and D06P (textiles dyeing or printing), which are usually combined with other technologies of strong symbiosis, so when these dependent technologies make certain breakthroughs, they will drive the rapid development of related technologies.

## 6. Discussion

The results of this paper are helpful in sorting out the competitive situation and potential technical opportunities of global textile new material technology R&D. Although the existing research objects in the academic community involve many emerging industries and technologies, there is no research on the global competitive situation and technology opportunity analysis of new textile materials. Therefore, this paper attempts to supplement the existing research in terms of research object and research content, to potentially fill the gap of existing research.

### 6.1. Regional Innovation Trends

The global textile new materials technology innovation is developing rapidly in the direction of high performance, green, intelligence and industrial chain synergy, and around the concept of "revolutionary fibers and textiles", it further integrates interdisciplinary achievements such as nanotechnology and bioengineering to promote the breakthrough of material performance limits. At the same time, digitalization and greening go hand in hand, reshaping the global textile value chain and realizing the transition from "manufacturing" to "intelligent manufacturing". To maintain its global leadership in the field of new materials technology, the United States has set a series of development goals at the comprehensive strategic planning level, including the National Nanotechnology Initiative (NNI, launched in 2000), the Materials Genome Project (launched in 2011), and the National Manufacturing Innovation Network (now known as Manufacturing America) in 2012. It has taken the lead in completing the layout of patented technology research and development and technology opportunity development in the world. Global companies in the chemical industry, advanced materials production, and materials science, represented by Procter & Gamble, Kimberley-Clark Global Ltd., and Dow Global Technologies, provide an important guarantee for the United States to maintain its global leadership position in the development of the new textile materials industry. In addition, Japan's textile new material technology research and development strength is also at the forefront of the world, as early as the 80s and 90s of the last century, the Japanese government began to take a series of measures to promote the development of new material technology, put forward the goal of "focusing on the practicability of new materials, considering the coordinated development of the environment and resources", so it occupies a

pivotal international position in the field of cutting-edge materials such as global carbon fiber technology, functional chemical products, new chemical materials, and man-made fibers.

Under the model of full-staff collaboration and innovation that promotes the reform and reorganization of large enterprise groups and encourages the integration of the public and private sectors, South Korea has become an important force in the technical fields of spinning, printing and dyeing, garment production, and textile machinery manufacturing. Since 2012, the government has established policies such as the Nano Convergence 2020 Project, the Third Basic Plan for Science and Technology, and the Korea Future Growth Engines Plan, creating global aramid fiber and new chemical material manufacturers represented by Kolon Industries and LG Chem, which have maintained Korea's global competitiveness in new textile material technology. The development of new textile materials in Europe is similar to that of the United States and Japan, and it attaches great importance to the construction of industrial clusters with its characteristics, establishing and improving the industrial standardization system based on core technology and core manufacturing, and with the intensification of global sustainable development and international trade competition, its new textile material innovation model focuses on industrial demand, emphasizing the added value of textile materials. In 2011, six major technologies, including advanced materials, were identified as key enabling technologies (KETs) for the EU industry, and materials and nanomaterial technology were funded and deployed as important research areas, among which textile new material technology occupies an increasingly important position in the field of EU science and technology development.

## 6.2. Challenges in China

Although China's textile new material technology R&D started relatively late, it has developed relatively rapidly, and has produced several technological achievements that have reached or approached the international advanced level. However, it still faces multiple challenges, mainly involving technology, industry, market and environment. Firstly, in the R&D and application of key technologies and equipment such as new fiber materials, advanced textile products, green manufacturing, and intelligent manufacturing, there is a relatively large gap with developed countries such as the United States and Japan, and the problem of "big but not strong, big but not excellent" is still more prominent, and it is hoped that the country will pay attention to the coherent support and long-term layout of the policy, as well as strengthen international technological innovation and cooperation, encourage domestic enterprises to pay attention to the core patent layout and technological innovation, and actively participate in international competition to enhance the competitiveness of enterprise products and technologies. Secondly, the low conversion rate of achievements of universities and scientific research institutes, the lack of R&D investment of enterprises and other industrial chain coordination and industrialization problems, the dependence on imports of high-end textile equipment and testing instruments, and the pressure of environmental protection and sustainable development restricts the overall upgrading of the whole industrial chain. Finally, the performance evaluation of new materials and the integration of testing standards with international standards need to be further improved, which will affect the market promotion and the competition pattern of application scenarios in emerging fields such as aerospace, defense, and medical.

## 6.3. Actionable recommendations

In view of the characteristics of industrial base, resource endowment and policy orientation in different regions, it is suggested that the advantages of each other, and the relevant and feasible R&D strategies for global textile new materials technology are as follows:

(1) In China, promote cooperation between universities and leading enterprises, focusing on the R&D and innovation of high-performance fibers, green materials and degradable textiles. (2) In Japan, we will strengthen the role of Toray, Teijin and other enterprises in technological refinement and high-end material leadership, establish a circular industrial chain of chemical fiber, textile, and consumer brands, and realize sustainable innovation in the circular economy. (3) In South Korea, vigorously develop the application of smart textiles and consumption scenarios, and expand the commercial market of functional materials for international companies such as Samsung Electronics and LG Chem. (4) In the United States, promote disruptive technology incubation and interdisciplinary collaborative development and innovation, promote the application and breakthrough of bioengineering materials in the textile field, and establish a traceability system for textile raw materials based on the blockchain. (5) In Europe, we will continue to strengthen the construction of sustainable standards and recycling systems, promote globally recognized standardization and certification systems, and realize the industrial cluster effect of regional industries such as high-end textiles in Italy, machinery manufacturing in Germany and bio-based materials in Northern Europe.

In summary, with the impact of COVID-19 (Corona Virus Disease 2019) and global geopolitics, countries are paying more and more attention to upgrading supply chain security, so they continue to increase national-level strategic support and investment and begin to promote the reform and restructuring of large enterprises to play a centralized role. Although the United States and Japan still have obvious advantages, the technology gap between various regions is narrowing, and the competition in the field of new textile material technology has entered a white-hot stage.

## 7. Conclusions and recommendations

### 7.1. Research summary

Based on the data of 94,525 invention patents for new textile material technology in the world, this paper analyzes the competitive situation and technical opportunities of technology R&D in this field, and finds that:

(1) Japanese companies are particularly prominent among the world's major patent holders, and internationally renowned multinational companies represented by Toray Corporation have the highest number of patents for new textile material technology, reflecting their very important international position in the field of fibers, functional chemical products, carbon fiber composite materials and other cutting-edge materials. The United States, China, and South Korea followed, and together they constitute the main body of innovation in the world's new textile material technology.

(2) In terms of technology layout, the United States and Japan are more forward-looking than other countries and regions. Kao Co., Ltd. and Kimberley-Clark Global Co., Ltd. began patent applications for new textile material technology in 2001, and Procter & Gamble and Kuraray Co., Ltd. also began to deploy in 2002. However, with the support of government policies and the integration of resources, the two countries' innovative entities of China and South Korea have developed rapidly, especially Kolon Industries, which applied for 18 patents in 2004, reflecting its strong innovation strength and competitiveness in this field, and has become an important development force for new textile material technology. Combined with the frequency of patent citations, the number of patents in the same family, and the number of countries, the study also found that the core patented technology of the United States in the field of new textile material technology is in a global

leading position, and has a strong sense of overseas layout, showing its characteristics of "big and strong, many and excellent" in the field of new textile material technology.

(3) the technical opportunities that countries around the world are relatively more concerned about mainly include D01F, D06M, D04H, and D01D, etc., indicating that the chemical characteristics of the production of artificial filament, the chemical characteristics of the fiber, the equipment dedicated to the production of carbon fiber, the manufacturing of textiles and the mechanical methods or equipment for making chemical filament and fiber are more favored by the market.

## 7.2. Limitations and future research

After decades of development, the global textile new material technology R&D pattern is changing, but the development of innovation bodies in various countries is unbalanced, and China and Japan are becoming the core forces of cooperative R&D in this field. Under the guidance of global sustainable development goals such as green, low-carbon, and energy-saving, new textile material technology needs to continue to strengthen diffusion, integration and alliance, and continue to play a leading role in international cooperation and technological innovation. Although this paper has reference significance and value for the research on filling the competitive situation and technical opportunities of global textile new material technology R&D, there are still some limitations: due to the shortcomings of the patent database, it is impossible to use the function of the technology roadmap to analyze the infrastructure and path of technology R&D evolution in more depth. At the same time, by limiting the scope of the study to the IP5, the research horizon will be subject to some limitations.

In summary, China's huge textile market demand and technology application, have played a positive role in promoting domestic textile technology innovation. As an essential part of the national industry-university-research collaborative innovation ecosystem, universities play an important role in the field of textile technology innovation in China, as well as in technology research and development and sustainable industrial development. Although the domestic textile new material technology research and development has achieved relatively rich results, the core patented technology and the layout of cutting-edge new materials are still facing huge challenges. How to solve the contradiction of "big but not strong, many but not excellent" in China is currently facing new challenges in China's innovation and development, technical planning, and layout in the textile industry.

Although this paper has reference significance and value for the research on the competitive situation and technical opportunities of global new textile materials technology R&D, there are some limitations. Due to limitations of the patent database and IP5 scope, which may not capture all forms of technology R&D collaboration, as some collaborations may not result in patent applications. In the future, more complete global data will be available to compare countries for further and more comprehensive analyses.

## Acknowledgments

We are grateful for all those who participated in the innovation management team of the Key Laboratory of Intelligent Textile and Flexible Inter-connection of Zhejiang Province and Zhejiang Sci-Tech University.

## Author contributions

**Conceptualization:** Pengfei Wang, Hua Cheng.

**Data curation:** Pengfei Wang.

**Formal analysis:** Pengfei Wang.

**Funding acquisition:** Pengfei Wang.

**Methodology:** Hua Cheng.

**Resources:** Pengfei Wang.

**Software:** Pengfei Wang.

**Supervision:** Hua Cheng.

**Validation:** Hua Cheng.

**Visualization:** Pengfei Wang.

**Writing – original draft:** Pengfei Wang.

**Writing – review & editing:** Pengfei Wang.

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
