## [Decision Letter · Decision Letter 0]

13 Aug 2024

PONE-D-24-23639Research on the competitive situation and technical opportunities of global new textile materials technology R&DPLOS ONE

Dear Dr. wang,

Thank you for submitting your manuscript to PLOS ONE. After careful consideration, we feel that it has merit but does not fully meet PLOS ONE’s publication criteria as it currently stands. Therefore, we invite you to submit a revised version of the manuscript that addresses the points raised during the review process.

The self explainatory comments of reviewers are there for your guidance. ==============================

We look forward to receiving your revised manuscript.

Kind regards,

Yasir Nawab, PhD

Academic Editor

PLOS ONE

 [This research was funded by The Funding of Key Laboratory of Intelligent Textile and Flexible Interconnection of Zhejiang Province, grant number 111504A4E22004, and Silk & Fashion Culture Research Center of Zhejiang Province, Zhejiang Provincial Key Research Institute of Philosophy and Social Science, grant number 2022JDKTYB27.].  

Additional Editor Comments:

Reviewers' comments:

Reviewer's Responses to Questions

**Comments to the Author**

1. Is the manuscript technically sound, and do the data support the conclusions?

Reviewer #1: Yes

Reviewer #2: Partly

2. Has the statistical analysis been performed appropriately and rigorously? 

Reviewer #1: Yes

Reviewer #2: Yes

3. Have the authors made all data underlying the findings in their manuscript fully available?

Reviewer #1: Yes

Reviewer #2: Yes

4. Is the manuscript presented in an intelligible fashion and written in standard English?

Reviewer #1: Yes

Reviewer #2: Yes

5. Review Comments to the Author

Reviewer #1: In the manuscript titled “Research on the competitive situation and technical opportunities of global new textile materials technology R&D” the researcher took the 94,525 invention patents for new textile material technology in the world as the research object and explored the competitive situation, technical opportunities, and layout characteristics of technology research and development in this field. .

Here are some points in the current study that need to be addressed for a better understanding of the research.

1. What was the criteria of selecting the patents to be added in this study? And how you have collected the data? Please explain the research methodology in detail.

2. What is meant by “D03 weaving, D04 weaving”, what is the difference between both? Please also explain the term D first then use the 1,2 and 3 etc.

3. The abbreviations are not explained in the text clearly, specifically used in table 4. Please explain.

4. You never mentioned that the patent includes local patents, or these are only international patents?

5. The literature review should be improved. Some other important studies must be added to this research. Below some important studies are referred to enrich the literature review.

• A review of artificial intelligence applications in apparel industry

(DOI: 10.1080/00405000.2021.1880088)

Reviewer #2: The article titled aims to analyze the global landscape of research and development (R&D) in new textile materials technology. It focuses on 94,525 invention patents related to new textile materials, examining the competitive positions, technical opportunities, and R&D layout characteristics. Although the article presents a valuable analysis of the global competitive situation and technical opportunities in new textile materials technology R&D, there are some critical issues and some suggestions for improvement:

• The abstract mentions the analysis of 94,525 invention patents but does not describe the specific methodology used for this analysis

• The abstract should succinctly point out the novel aspects of the research, and main contributions to the literature or practice

• While the introduction sets the context and significance of the study, it fails to explicitly state the research gap

• The introduction does not clearly outline the research methods used in the study. This is a critical component as it provides readers with an understanding of how the research was conducted and ensures transparency. It does not mention similar studies or previous research in the field

• The introduction should clearly define the research objectives. While the goals are somewhat implied, a concise statement of the specific objectives would enhance clarity. Also, including a clear hypothesis or specific research questions can provide a focused direction for the study and make it easier for readers to understand what the research aims to investigate

• The introduction does not provide an overview of the structure of the article. Mentioning the structure helps guide the reader through the paper, making it easier to follow the argument and understand the flow of the research

• The literature review, while well-structured, offers a broad range of research areas but lacks in-depth analysis of existing studies. Specifically, it could benefit from a more rigorous discussion of research on competitive situations, technical opportunities in new material technology, and policy mechanisms for innovation in new textile materials.

• The methodology section is well-defined and detailed, ensuring that the research method is transparent and replicable. However, the abstract and introduction could be improved by briefly mentioning the research method to give readers a more comprehensive overview from the outset.

• To effectively communicate the connection between patent analysis and policy implications, the study should provide a clearer explanation of how the trends of countries and policy recommendations were drawn from the patent data analysis

• articulate the novel ideas, concepts, or frameworks you introduce to the existing body of knowledge within the field. What are the theoretical contributions of the paper? What are its practical contributions?

6. PLOS authors have the option to publish the peer review history of their article (what does this mean? ). If published, this will include your full peer review and any attached files.

**Do you want your identity to be public for this peer review?** For information about this choice, including consent withdrawal, please see our Privacy Policy .

Reviewer #1: **Yes: ** Raja Muhammad Waseem Ullah Khan

Reviewer #2: **Yes: ** Sina Tarighi

---

## [Author Response · Author response to Decision Letter 1]

12 Sep 2024

Dear Reviewers

We are truly grateful to yours and other reviewers’ critical comments and thoughtful suggestions on our manuscript (Manuscript ID: PONE-D-24-23639). We have revised the manuscript and responded to these questions point-by-point based on the comments and suggestions. Additionally, the list of changes is given below. Now we are submitting this revised version after making improvements. If you have any other questions about this manuscript, please do not hesitate to contact me.

Reviewer #1: In the manuscript titled “Research on the competitive situation and technical opportunities of global new textile materials technology R&D” the researcher took the 94,525 invention patents for new textile material technology in the world as the research object and explored the competitive situation, technical opportunities, and layout characteristics of technology research and development in this field. .

Here are some points in the current study that need to be addressed for a better understanding of the research.

1. What was the criteria of selecting the patents to be added in this study? And how you have collected the data? Please explain the research methodology in detail.

Response 1: Thanks for your suggestions, the data used in this article are from the PatSnap patent database (https:// analytics.zhihuiya.com/). As for new textile material technology, according to the research results of Slubik et al. , Papaspyrides et al. , and Sibiescu et al. , it is mainly defined as high-tech fibers, high-tech fabrics, and high-tech industrial textiles produced by adopting high-tech processes and equipment in recent years. Its content involves textiles, clothing, fibers, spinning, weaving and knitting, etc. The International Patent Classification (IPC) includes two categories: D (textile) and A41 (garment). First you select the new material industry in the national strategic emerging industry classification, and combine it with the international patent classification number D01 fiber, D02 yarn, D03 weaving, D04 knitting, D05 sewing, and D06 fabric treatment, etc.

The process of collecting the data was as follows: firstly, select the patent receiving offices in 170 countries and regions around the world provided by the PatSnap system, and then download the data that meets the requirements according to the criteria. Statistical analysis is carried out by patentometrics to obtain the data of major patentees, technical field and core patents, and finally, the technical opportunities of new textile materials in the world are analyzed through the model.

2. What is meant by “D03 weaving, D04 weaving”, what is the difference between both? Please also explain the term D first then use the 1,2 and 3 etc.

Response 2:Thanks for your suggestions, we're so sorry that D03(weaving) and D04(knitting) are not the same, it’s our mistake, we've revised in the manuscript. According to the International Patent Classification(IPC), the technical fields related to patents are divided into eight parts: A, B, C, D, E, F, G, and H. Textile technology, together with papermaking technology, is in the Part D, including D01-D07. D01-D06 represents the aspects of raw materials, spinning, weaving, knitting, sewing and fabric treatment. Please see the official reference URL, https://www.wipo.int/classifications/ipc/en/.

3. The abbreviations are not explained in the text clearly, specifically used in table 4. Please explain.

Response 3: Thanks for your suggestions, we have added explanations and descriptions of abbreviations in the manuscript, in particular in Table 4, and highlighted them in different colours.

4. You never mentioned that the patent includes local patents, or these are only international patents?

Response 4: Thanks for your suggestions, we selected the patent receiving offices in 170 countries and regions around the world provided by the PatSnap system, and data includes both local patents and international patents.

5. The literature review should be improved. Some other important studies must be added to this research. Below some important studies are referred to enrich the literature review.

• A review of artificial intelligence applications in apparel industry

(DOI: 10.1080/00405000.2021.1880088)

Response 5: Thanks for your suggestions, we have enriched the literature review by adding some other important studies, including references to some important studies recommended by reviewers, and the revisions have been highlighted in different colours.

Reviewer #2: The article titled aims to analyze the global landscape of research and development (R&D) in new textile materials technology. It focuses on 94,525 invention patents related to new textile materials, examining the competitive positions, technical opportunities, and R&D layout characteristics. Although the article presents a valuable analysis of the global competitive situation and technical opportunities in new textile materials technology R&D, there are some critical issues and some suggestions for improvement:

• The abstract mentions the analysis of 94,525 invention patents but does not describe the specific methodology used for this analysis

Response 1: Thanks for your suggestions, we've added the specific methodology used in the abstract, and the changes have been highlighted in different colors. It reads as follows:

This paper takes the 94,525 invention patents for new textile material technology in the world as the research object,and uses competitive situation calculation and patentometrics to explore the competitive situation, technical opportunities,and layout characteristics of technology R&D (Research and Development) in this field.

• The abstract should succinctly point out the novel aspects of the research, and main contributions to the literature or practice

Response 2: Thanks for your suggestions, we've added the novel aspects of the research and main contributions to the literature or practice in the abstract, and the changes have been highlighted in different colors. It reads as follows:

This study effectively supplement the theoretical analysis of competitive situation and technical opportunities of technology R&D, and enrich and expand the empirical research on international new textile materials technology and innovation.

• While the introduction sets the context and significance of the study, it fails to explicitly state the research gap

Response 3: Thanks for your suggestions, we've added the research gap in the introduction, and the changes have been highlighted in different colors. It reads as follows:

However, there is a lack of research of the competitive situation and technical opportunities of global new textile materials technology R&D, and the empirical study on international new textile materials technology and innovation can provide a more accurate and comprehensive panoramic analysis, serving a reference for the government's science and technology innovation policy formulation, technology layout and R&D. Therefore, this paper attempts to supplement the existing research in terms of research object and research content, to potentially fill the gap of existing research.

• The introduction does not clearly outline the research methods used in the study. This is a critical component as it provides readers with an understanding of how the research was conducted and ensures transparency. It does not mention similar studies or previous research in the field

Response 4: Thanks for your suggestions, we've added the research methods used in the study, and mentioned similar studies or previous research in the field. The changes have been highlighted in different colors. It reads as follows:

Patent measurement is a method to comprehensively understand the level,the new emerging clustering technique and trend of industrial technology development, and technology measurement analysis with patentometrics as the main method is one of the effective and practical methods,including examining the innovation activities of firms in the textile and clothing industry[4-6].

• The introduction should clearly define the research objectives. While the goals are somewhat implied, a concise statement of the specific objectives would enhance clarity. Also, including a clear hypothesis or specific research questions can provide a focused direction for the study and make it easier for readers to understand what the research aims to investigate

Response 5: Thanks for your suggestions, we have added specific research questions, and provide a focused direction for the study in the introduction. However, due to the limited space of the manuscript, we're so sorry that we are unable to provide more content.

• The introduction does not provide an overview of the structure of the article. Mentioning the structure helps guide the reader through the paper, making it easier to follow the argument and understand the flow of the research

Response 6: Thanks for your suggestions, the structure of the paper is arranged as follows: Literature Review in Section 2, data sources in Section 3, methods in Section 4, results of empirical analysis in Section 5, discussion and conclusions in Section 6, and finally limitations and future research in Section 7.

• The literature review, while well-structured, offers a broad range of research areas but lacks in-depth analysis of existing studies. Specifically, it could benefit from a more rigorous discussion of research on competitive situations, technical opportunities in new material technology, and policy mechanisms for innovation in new textile materials.

Response 7: Thanks for your suggestions, however, due to the limited space of the manuscript, we're so sorry that we are unable to provide more content.

• The methodology section is well-defined and detailed, ensuring that the research method is transparent and replicable. However, the abstract and introduction could be improved by briefly mentioning the research method to give readers a more comprehensive overview from the outset.

Response 8: Thanks for your suggestions, we have added more content in the abstract and introduction, which give readers a more comprehensive overview from the outset by briefly mentioning the research method.

• To effectively communicate the connection between patent analysis and policy implications, the study should provide a clearer explanation of how the trends of countries and policy recommendations were drawn from the patent data analysis

Response 9: Thanks for your suggestions, We have elaborated on the connection between patent analysis and policy implications in the discussion and conclusions of Section 6. However, due to the limited space of the manuscript, we're so sorry that we are unable to provide more content.

• articulate the novel ideas, concepts, or frameworks you introduce to the existing body of knowledge within the field. What are the theoretical contributions of the paper? What are its practical contributions?

Response 10: Thanks for your suggestions, we've added the theoretical contributions and practical contributions in the introduction. The changes have been highlighted in different colors. It reads as follows:

The possible marginal contributions of this paper are as follows: (1) Supplementation of the research framework. Based on the 94,525 invention patents for new textile material technology in the world as the research object, this paper constructs the the competitive situation, technical opportunities,and layout characteristics of 170 countries and regions around the world patent technology innovation in order to effectively supplement the analytical framework of international technology R&D. (2) Expansion of research methods. This paper uses competitive situation calculation and patentometrics to scientifically explore the characteristic factors of technology R&D ability and technical opportunities so as to enrich and expand the empirical research on international technology innovation and provide policy enlightenment for optimizing regional sustainable development and national R&D innovation.

---

## [Decision Letter · Decision Letter 1]

12 Nov 2024

PONE-D-24-23639R1Research on the competitive situation and technical opportunities of global new textile materials technology R&DPLOS ONE

Dear Dr. wang,

Thank you for submitting your manuscript to PLOS ONE. After careful consideration, we feel that it has merit but does not fully meet PLOS ONE’s publication criteria as it currently stands. Therefore, we invite you to submit a revised version of the manuscript that addresses the points raised during the review process.

We look forward to receiving your revised manuscript.

Kind regards,

Zeashan Hameed Khan, Ph.D.

Academic Editor

PLOS ONE

Additional Editor Comments:

In the revised version, authors have improved the major gaps existing in the first submission. However, the reply to the reviewer's queries are not found satisfactory. Please take it seriously and do not worry about the additional length of the paper in case the explanation is required to be added.

Reviewers' comments:

Reviewer's Responses to Questions

**Comments to the Author**

1. If the authors have adequately addressed your comments raised in a previous round of review and you feel that this manuscript is now acceptable for publication, you may indicate that here to bypass the “Comments to the Author” section, enter your conflict of interest statement in the “Confidential to Editor” section, and submit your "Accept" recommendation.

Reviewer #1: All comments have been addressed

Reviewer #2: (No Response)

2. Is the manuscript technically sound, and do the data support the conclusions?

Reviewer #1: Yes

Reviewer #2: Partly

3. Has the statistical analysis been performed appropriately and rigorously? 

Reviewer #1: N/A

Reviewer #2: Yes

4. Have the authors made all data underlying the findings in their manuscript fully available?

Reviewer #1: Yes

Reviewer #2: Yes

5. Is the manuscript presented in an intelligible fashion and written in standard English?

Reviewer #1: Yes

Reviewer #2: Yes

6. Review Comments to the Author

Reviewer #1: (No Response)

Reviewer #2: The authors have made improvements to certain sections of the manuscript based on the comments provided. However, it is not acceptable to claim that additional content cannot be added due to "space limitations", especially in response to major comments. Therefore:

A significant issue that remains unaddressed is the need to communicate the connection between patent analysis and its policy implications. The authors should provide a clearer explanation of how the trends observed in different countries were derived from the patent data analysis, as well as the basis for the policy recommendations suggested. Without a clear understanding of the mechanisms involved, readers will struggle to engage with the results.

It is essential that the authors offer a more thorough explanation of the theoretical foundations and their interconnections, even briefly.

Additionally, I suggest that the "data sources" section can be placed under the "methods" category for better coherence and organization

7. PLOS authors have the option to publish the peer review history of their article (what does this mean? ). If published, this will include your full peer review and any attached files.

**Do you want your identity to be public for this peer review?** For information about this choice, including consent withdrawal, please see our Privacy Policy .

Reviewer #1: No

Reviewer #2: **Yes: ** sina tarighi

---

## [Author Response · Author response to Decision Letter 2]

12 Jan 2025

Dear Reviewers

We are truly grateful to yours and other reviewers’ critical comments and thoughtful suggestions on our manuscript (Manuscript ID: PONE-D-24-23639). We have revised the manuscript and responded to these questions point-by-point based on the comments and suggestions. Additionally, the list of changes is given below. Now we are submitting this revised version after making improvements. If you have any other questions about this manuscript, please do not hesitate to contact me.

1. A significant issue that remains unaddressed is the need to communicate the connection between patent analysis and its policy implications. The authors should provide a clearer explanation of how the trends observed in different countries were derived from the patent data analysis, as well as the basis for the policy recommendations suggested. Without a clear understanding of the mechanisms involved, readers will struggle to engage with the results.

Response 1: Thanks for your suggestions, drawing on the existing research results, we put the mechanism explanation and the basis for the policy recommendations suggested in the theoretical analysis part of the paper.. It reads as follows:

In recent years, technology competition in emerging industries has become increasingly fierce, and more and more scholars have identified the main competitors in specific technology fields from the perspective of patent competition analysis or patent cooperation analysis, and helped innovation subjects to construct patent strategies and plans to achieve the goal of enhancing competitive advantage by tracking the trend of technology development and evaluating the innovation ability of technology[32].With the rapid development of digital technology, the innovation ability of enterprises has become an important measure of national competitiveness. Therefore, the policy influence of different countries and the research suggestions of scholars have also had a positive effect on the decisive factors for innovative subjects to gain competitive advantages in the market, and provide an important reference for the country to formulate relevant technological strategies and promote the transformation of scientific and technological achievements[33].

At the same time, as the premise of technology R&D and innovation, technology opportunity research has become a key issue of concern to the state and enterprises, and has important strategic significance for technology competition, market layout, regional innovation and industrial development. Therefore, mining technological opportunities from patent data can not only obtain possible new technologies, or new technologies that are improved on the basis of existing technologies, but also provide enterprises with important R&D intelligence and reference directions through analysis[34-36].

2. It is essential that the authors offer a more thorough explanation of the theoretical foundations and their interconnections, even briefly.

Response 2:Thanks for your suggestions, we have added a separate section of theoretical analysis to the paper, focusing on explanation of the theoretical foundations and their interconnections. It reads as follows:

Theoretical analysis

Patent data mining is the use of patent analysis to extract valuable data information from patent documents, such as the state of industrial technology development, the main areas of technological competition and competitors' technology layout. Through patent competitive intelligence, it helps enterprises understand the industry environment, evaluate themselves and monitor their competitors so as to obtain competitive advantages and potential opportunities[29-31].

In recent years, technology competition in emerging industries has become increasingly fierce, and more and more scholars have identified the main competitors in specific technology fields from the perspective of patent competition analysis or patent cooperation analysis, and helped innovation subjects to construct patent strategies and plans to achieve the goal of enhancing competitive advantage by tracking the trend of technology development and evaluating the innovation ability of technology[32].With the rapid development of digital technology, the innovation ability of enterprises has become an important measure of national competitiveness. Therefore, the policy influence of different countries and the research suggestions of scholars have also had a positive effect on the decisive factors for innovative subjects to gain competitive advantages in the market, and provide an important reference for the country to formulate relevant technological strategies and promote the transformation of scientific and technological achievements[33].

At the same time, as the premise of technology R&D and innovation, technology opportunity research has become a key issue of concern to the state and enterprises, and has important strategic significance for technology competition, market layout, regional innovation and industrial development. Therefore, mining technological opportunities from patent data can not only obtain possible new technologies, or new technologies that are improved on the basis of existing technologies, but also provide enterprises with important R&D intelligence and reference directions through analysis[34-36].

To sum up, the continuous innovation of technology will promote the development of the industry, this process on the one hand requires the progress of technology itself, on the other hand, there is also a demand from the market, the two need to reach a technical research and development in the unity of technical and economic possibilities[37]. Therefore, on the basis of existing research, this paper analyzes the competitive situation and technical opportunities of global textile new material technology research and development in the 21st century, in order to provide a useful reference for the forward-looking and strategic research and development layout of the government, industry and enterprise technology innovation.

3. Additionally, I suggest that the "data sources" section can be placed under the "methods" category for better coherence and organization .

Response 3: Thanks for your suggestions, we have placed the "data sources" section under the "methods" category and have moderately adjusted the structure of the paper.

---

## [Decision Letter · Decision Letter 2]

31 Jan 2025

PONE-D-24-23639R2Research on the competitive situation and technical opportunities of global new textile materials technology R&DPLOS ONE

Dear Dr. wang,

Thank you for submitting your manuscript to PLOS ONE. After careful consideration, we feel that it has merit but does not fully meet PLOS ONE’s publication criteria as it currently stands. Therefore, we invite you to submit a revised version of the manuscript that addresses the points raised during the review process.

We look forward to receiving your revised manuscript.

Kind regards,

Zeashan Hameed Khan, Ph.D.

Academic Editor

PLOS ONE

Journal Requirements:

Additional Editor Comments:

The paper has been improved and it can be accepted after minor improvements. Please look at the reviewer's comments carefully. All abbreviations must be explained at their first use.

Figures and Tables must be referenced in the text. Check all references. They must follow the Plos One style with complete information.

Reviewers' comments:

Reviewer's Responses to Questions

**Comments to the Author**

1. If the authors have adequately addressed your comments raised in a previous round of review and you feel that this manuscript is now acceptable for publication, you may indicate that here to bypass the “Comments to the Author” section, enter your conflict of interest statement in the “Confidential to Editor” section, and submit your "Accept" recommendation.

Reviewer #1: (No Response)

Reviewer #2: (No Response)

2. Is the manuscript technically sound, and do the data support the conclusions?

Reviewer #1: Yes

Reviewer #2: Partly

3. Has the statistical analysis been performed appropriately and rigorously? 

Reviewer #1: N/A

Reviewer #2: Yes

4. Have the authors made all data underlying the findings in their manuscript fully available?

Reviewer #1: Yes

Reviewer #2: Yes

5. Is the manuscript presented in an intelligible fashion and written in standard English?

Reviewer #1: Yes

Reviewer #2: Yes

6. Review Comments to the Author

Reviewer #1: The manuscript titled " Research on the competitive situation and technical opportunities of global new textile materials technology R&D" provides valuable insights into the competitive landscape and technical opportunities in the field of textile material innovation. By analyzing 94,525 patents, the study identifies key contributors, technological areas, and trends across major regions, highlighting the global R&D dynamics.

• While the manuscript is rich in data and offers a global perspective, it requires improvements in structure, clarity, and data presentation.

• Additionally, expanding the discussion on limitations and contextualizing findings within existing literature will strengthen its academic contribution.

• Overall, the manuscript has significant potential but requires minor revisions to fully realize its impact.

Here are some points in the current study that need to be addressed for a better understanding of the research.

1. Break the discussion section into sub-sections with clear headings (e.g., "Regional Innovation Trends," "Challenges in China") for better readability.

2. Elaborate on the limitations of the patent database and IP5 scope, explaining how these might affect the comprehensiveness of the findings.

3. Compare the findings with existing studies on textile material innovation to position the work within the broader academic discourse.

4. Revise lengthy and complex sentences to make the manuscript more accessible to a wider audience.

5. Add a subsection with actionable recommendations tailored to different regions or stakeholders.

6. Summarize major findings from the discussion more concisely to avoid repetitive or verbose descriptions.

7. Ensure proper citation of all patent data, government policies, and industrial initiatives mentioned in the manuscript.

8. Emphasize how the findings can inform industry practices, R&D strategies, and policy frameworks.

9. The literature review should be improved. Some other important studies must be added to this research. Below are some important studies that can enrich literature review.

•

• Comparative Geometrical Analysis of In Situ Mechanical Performance of 2-D Woven In-Plane Auxetic Structures (DOI: 10.1520/JTE20230195)

• Numerical Analysis of Binding Yarn Float Length for 3D Auxetic Structures (DOI: 10.1002/pssb.202000440)

• Influence of tetrahedral architectures on fluid transmission and heat retention behaviors of auxetic weaves (DOI: 10.1016/j.tsep.2023.101946)

• A review of artificial intelligence applications in apparel industry (DOI: 10.1080/00405000.2021.1880088)

• Study of comfort performance of novel car seat design for long drive (DOI: 10.1177/0954407019852279)

• Cellulosic Fillers Extracted from Argyreia Speciose Waste: A Potential Reinforcement for Composites to Enhance Properties (DOI:10.1080/15440478.2020.1856271)

Reviewer #2: I appreciate the effort and responsibility you have demonstrated in addressing the previous comments. Compared to the earlier revision, this version reflects a more thoughtful attempt to clarify key aspects of the study.

However, there remains one crucial issue that still needs to be addressed. My expectation was not just a general theoretical discussion on the link between patent analysis and policy recommendations, but rather a clear and explicit explanation of how the specific findings of your study directly lead to the policy recommendations you propose. In other words, while you have articulated the broader theoretical foundation, the manuscript still lacks a concrete connection between the empirical results and the suggested policies in the conclusion.

This connection is essential for ensuring the validity and impact of your conclusions. Without it, the policy implications may appear disconnected from the actual findings, which could raise concerns about the robustness of the study’s contributions. I strongly encourage you to make this link explicit by demonstrating how the trends and patterns observed in your patent analysis serve as the basis for the policy recommendations you suggest.

I look forward to reviewing the revised version and appreciate your efforts in further refining the manuscript.

7. PLOS authors have the option to publish the peer review history of their article (what does this mean? ). If published, this will include your full peer review and any attached files.

**Do you want your identity to be public for this peer review?** For information about this choice, including consent withdrawal, please see our Privacy Policy .

Reviewer #1: No

Reviewer #2: **Yes: ** sina tarighi

---

## [Author Response · Author response to Decision Letter 3]

10 Mar 2025

PLOS ONE Decision: Revision required [PONE-D-24-23639] –

Dear Reviewers

We are truly grateful to yours and other reviewers’ critical comments and thoughtful suggestions on our manuscript (Manuscript ID: PONE-D-24-23639). We have revised the manuscript and responded to these questions point-by-point based on the comments and suggestions. Additionally, the list of changes is given below. Now we are submitting this revised version after making improvements. If you have any other questions about this manuscript, please do not hesitate to contact me.

Reviewer #1:

1. Break the discussion section into sub-sections with clear headings (e.g., "Regional Innovation Trends," "Challenges in China") for better readability.

Response 1: Thanks for your suggestions, we have added the discussion section into sub-sections with clear headings, it reads below:

. 6. Discussion

The results of this paper are helpful in sorting out the competitive situation and potential technical opportunities of global textile new material technology R&D. Although the existing research objects in the academic community involve many emerging industries and technologies, there is no research on the global competitive situation and technology opportunity analysis of new textile materials. Therefore, this paper attempts to supplement the existing research in terms of research object and research content, to potentially fill the gap of existing research.

6.1 Regional Innovation Trends

The global textile new materials technology innovation is developing rapidly in the direction of high performance, green, intelligence and industrial chain synergy, and around the concept of "revolutionary fibers and textiles", it further integrates interdisciplinary achievements such as nanotechnology and bioengineering to promote the breakthrough of material performance limits. At the same time, digitalization and greening go hand in hand, reshaping the global textile value chain and realizing the transition from "manufacturing" to "intelligent manufacturing". To maintain its global leadership in the field of new materials technology, the United States has set a series of development goals at the comprehensive strategic planning level,including the National Nanotechnology Initiative ( NNI, launched in 2000 ), the Materials Genome Project ( launched in 2011 ), and the National Manufacturing Innovation Network (now known as Manufacturing America) in 2012. It has taken the lead in completing the layout of patented technology research and development and technology opportunity development in the world. Global companies in the chemical industry, advanced materials production, and materials science, represented by Procter & Gamble, Kimberley-Clark Global Ltd., and Dow Global Technologies, provide an important guarantee for the United States to maintain its global leadership position in the development of the new textile materials industry. In addition, Japan's textile new material technology research and development strength is also at the forefront of the world, as early as the 80s and 90s of the last century, the Japanese government began to take a series of measures to promote the development of new material technology, put forward the goal of "focusing on the practicability of new materials, considering the coordinated development of the environment and resources", so it occupies a pivotal international position in the field of cutting-edge materials such as global carbon fiber technology, functional chemical products, new chemical materials, and man-made fibers.

Under the model of full-staff collaboration and innovation that promotes the reform and reorganization of large enterprise groups and encourages the integration of the public and private sectors, South Korea has become an important force in the technical fields of spinning, printing and dyeing, garment production, and textile machinery manufacturing. Since 2012, the government has established policies such as the Nano Convergence 2020 Project, the Third Basic Plan for Science and Technology, and the Korea Future Growth Engines Plan, creating global aramid fiber and new chemical material manufacturers represented by Kolon Industries and LG Chem, which have maintained Korea's global competitiveness in new textile material technology. The development of new textile materials in Europe is similar to that of the United States and Japan, and it attaches great importance to the construction of industrial clusters with its characteristics, establishing and improving the industrial standardization system based on core technology and core manufacturing, and with the intensification of global sustainable development and international trade competition, its new textile material innovation model focuses on industrial demand, emphasizing the added value of textile materials. In 2011, six major technologies, including advanced materials, were identified as key enabling technologies (KETs) for the EU industry, and materials and nanomaterial technology were funded and deployed as important research areas, among which textile new material technology occupies an increasingly important position in the field of EU science and technology development.

6.2 Challenges in China

Although China's textile new material technology R&D started relatively late,it has developed relatively rapidly, and has produced several technological achievements that have reached or approached the international advanced level. However, it still faces multiple challenges, mainly involving technology, industry, market and environment. Firstly, in the R&D and application of key technologies and equipment such as new fiber materials, advanced textile products, green manufacturing, and intelligent manufacturing, there is a relatively large gap with developed countries such as the United States and Japan, and the problem of "big but not strong, big but not excellent" is still more prominent, and it is hoped that the country will pay attention to the coherent support and long-term layout of the policy, as well as strengthen international technological innovation and cooperation, encourage domestic enterprises to pay attention to the core patent layout and technological innovation, and actively participate in international competition to enhance the competitiveness of enterprise products and technologies. Secondly, the low conversion rate of achievements of universities and scientific research institutes, the lack of R&D investment of enterprises and other industrial chain coordination and industrialization problems, the dependence on imports of high-end textile equipment and testing instruments, and the pressure of environmental protection and sustainable development restricts the overall upgrading of the whole industrial chain. Finally, the performance evaluation of new materials and the integration of testing standards with international standards need to be further improved, which will affect the market promotion and the competition pattern of application scenarios in emerging fields such as aerospace, defense, and medical.

In summary, with the impact of COVID-19 (Corona Virus Disease 2019) and global geopolitics,countries are paying more and more attention to upgrading supply chain security, so they continue to increase national-level strategic support and investment and begin to promote the reform and restructuring of large enterprises to play a centralized role. Although the United States and Japan still have obvious advantages, the technology gap between various regions is narrowing, and the competition in the field of new textile material technology has entered a white-hot stage.

2. Elaborate on the limitations of the patent database and IP5 scope, explaining how these might affect the comprehensiveness of the findings.

Response 2: Thanks for your suggestions, we have added the limitation explaination of the patent database and IP5 scope, which reads as follows:

Although this paper has reference significance and value for the research on the competitive situation and technical opportunities of global new textile materials technology R&D, there are some limitations. Due to limitations of the patent database and IP5 scope, which may not capture all forms of technology R&D collaboration, as some collaborations may not result in patent applications. In the future,more complete global data will be available to compare countries for further and more comprehensive analyses.

3. Compare the findings with existing studies on textile material innovation to position the work within the broader academic discourse .

Response 3: Thanks for your suggestions, we have compared the findings with existing studies on textile material innovation, which reads as follows:

4. Revise lengthy and complex sentences to make the manuscript more accessible to a wider audience.

Response 4: Thanks for your suggestions, we have revised lengthy and complex sentences to make the manuscript more accessible to a wider audience, all changes are shown in different colors in the paper. .

5. Add a subsection with actionable recommendations tailored to different regions or stakeholders.

Response 5: Thanks for your suggestions, we have added a subsection with actionable recommendations tailored to different regions or stakeholders, which reads as follows:

6.3 Actionable recommendations

In view of the characteristics of industrial base, resource endowment and policy orientation in different regions, it is suggested that the advantages of each other, and the relevant and feasible R&D strategies for global textile new materials technology are as follows:

(1) In China, promote cooperation between universities and leading enterprises, focusing on the R&D and innovation of high-performance fibers, green materials and degradable textiles. (2) In Japan, we will strengthen the role of Toray, Teijin and other enterprises in technological refinement and high-end material leadership, establish a circular industrial chain of chemical fiber, textile, and consumer brands, and realize sustainable innovation in the circular economy. (3) In South Korea, vigorously develop the application of smart textiles and consumption scenarios, and expand the commercial market of functional materials for international companies such as Samsung Electronics and LG Chem. (4) In the United States, promote disruptive technology incubation and interdisciplinary collaborative development and innovation, promote the application and breakthrough of bioengineering materials in the textile field, and establish a traceability system for textile raw materials based on the blockchain. (5) In Europe, we will continue to strengthen the construction of sustainable standards and recycling systems, promote globally recognized standardization and certification systems, and realize the industrial cluster effect of regional industries such as high-end textiles in Italy, machinery manufacturing in Germany and bio-based materials in Northern Europe.

6. Summarize major findings from the discussion more concisely to avoid repetitive or verbose descriptions.

Response 6: Thanks for your suggestions, we have summarized major findings from the discussion more concisely to avoid repetitive or verbose descriptions, all changes are shown in different colors in the paper. .

7. Ensure proper citation of all patent data, government policies, and industrial initiatives mentioned in the manuscript.

Response 7: Thanks for your suggestions, we have ensured proper citation of all patent data, government policies, and industrial initiatives mentioned in the manuscript.

8. Emphasize how the findings can inform industry practices, R&D strategies, and policy frameworks.

Response 8: Thanks for your suggestions, we have emphasized how the findings can inform industry practices, R&D strategies, and policy frameworks, which reads as follows in the Introduction of paper:

The significance of this study is as follows: (1) Laying a theoretical foundation for future emerging technology cooperation in R&D and innovation. (2) Emphasize the findings can inform industry practices, R&D strategies, and policy frameworks. (3) Through the empirical analysis of global patent data, it provides a reference and basis for the government to rationally allocate innovation resources and formulate relevant policies, which has practical significance and application value for the decision-making of enterprise technology R&D departments.

9. The literature review should be improved. Some other important studies must be added to this research. Below are some important studies that can enrich literature review.

• Comparative Geometrical Analysis of In Situ Mechanical Performance of 2-D Woven In-Plane Auxetic Structures (DOI: 10.1520/JTE20230195)

• Numerical Analysis of Binding Yarn Float Length for 3D Auxetic Structures (DOI: 10.1002/pssb.202000440)

• Influence of tetrahedral architectures on fluid transmission and heat retention behaviors of auxetic weaves (DOI: 10.1016/j.tsep.2023.101946)

• A review of artificial intelligence applications in apparel industry (DOI: 10.1080/00405000.2021.1880088)

• Study of comfort performance of novel car seat design for long drive (DOI: 10.1177/0954407019852279)

• Cellulosic Fillers Extracted from Argyreia Speciose Waste: A Potential Reinforcement for Composites to Enhance Properties (DOI:10.1080/15440478.2020.1856271)

Response 9: Thanks for your suggestions, we have improved the literature review, which reads as follows:

2. Literature Review

At present, scholars' research on the technological innovation of new textile materials mainly focuses on three aspects: (1) In the field of textile raw material technology, such as herbal biomass, natural dyes, and extracted cellulose and fiber extraction[7], which are widely used in the development of high-performance, smart textiles, and multi-functional materials field[8, 9], which is to meet the needs of the industry and the market, and improve the added value and competitiveness of textiles. (2) Case studies focusing on technological innovations in new textile materials, such as the use of eco-friendly technologies to produce textiles with the required physicochemical, woven auxetic fabrics and mechanical properties field[10-12],as well as scalable nanofabrication technologies to tailor smart fabrics [13] and natural fabrics [14], to meet various potential application needs. (3) Analysis of the policy mechanism of technological innovation of new textile materials, evaluates the expected impact of new technologies on topics such as the environment, automotive, aerospace, and ballistic areas and human security[15-17], and emphasizes the important role of policy guidance and supervision in the sustainable development of technology. In addition, artificial intelligence (AI) technology is also gaining more and more attention in the apparel industries, and researchers are constantly exploring and optimizing the use of AI techniques in a range of industrial applications[18, 19]. Since patents are synonymous with innovative activities, the analysis of new textile material technology through patents has also attracted the attention of many scholars[20].

Core patents reflect the value of patented technologies, and the corresponding technologies or products can occupy market share and obtain high economic benefits by their competitiveness, so they have a high degree of technological competitive advantage and market economic value[21, 22].At present, academic research focuses more on the technical perspective, pointing out that the core patents are the technical patents that must be used to manufacture a certain technical field or a certain product without circumvention. Some scholars also believe that core patents are patents or patent portfolios with an important economic and strategic value that are in a key position of innovation, influential, and inevitable in a certain technical field[23-25].Based on this research, this paper defines core patents as patents that are in a fundamental position in the technological evolution route,absolutely leading and contributing to the subsequent development of the technology with a high reference value for epitaxial technology.

The research results on technological opportunit

---

## [Decision Letter · Decision Letter 3]

14 Mar 2025

Research on the competitive situation and technical opportunities of global new textile materials technology R&D

PONE-D-24-23639R3

Dear Dr. wang,

We’re pleased to inform you that your manuscript has been judged scientifically suitable for publication and will be formally accepted for publication once it meets all outstanding technical requirements.

Kind regards,

Zeashan Hameed Khan, Ph.D.

Academic Editor

PLOS ONE

Additional Editor Comments (optional):

The paper has been improved significantly and it can be accepted in the present form.

Reviewers' comments:

Reviewer's Responses to Questions

**Comments to the Author**

1. If the authors have adequately addressed your comments raised in a previous round of review and you feel that this manuscript is now acceptable for publication, you may indicate that here to bypass the “Comments to the Author” section, enter your conflict of interest statement in the “Confidential to Editor” section, and submit your "Accept" recommendation.

Reviewer #1: All comments have been addressed

Reviewer #2: All comments have been addressed

2. Is the manuscript technically sound, and do the data support the conclusions?

Reviewer #1: Yes

Reviewer #2: Yes

3. Has the statistical analysis been performed appropriately and rigorously? 

Reviewer #1: Yes

Reviewer #2: Yes

4. Have the authors made all data underlying the findings in their manuscript fully available?

Reviewer #1: Yes

Reviewer #2: Yes

5. Is the manuscript presented in an intelligible fashion and written in standard English?

Reviewer #1: Yes

Reviewer #2: Yes

6. Review Comments to the Author

Reviewer #1: I appreciate the effort and responsibility you have demonstrated in addressing the previous

comments. Compared to the earlier revision, this version reflects a more thoughtful attempt to

clarify key aspects of the study.

Reviewer #2: (No Response)

7. PLOS authors have the option to publish the peer review history of their article (what does this mean? ). If published, this will include your full peer review and any attached files.

**Do you want your identity to be public for this peer review?** For information about this choice, including consent withdrawal, please see our Privacy Policy .

Reviewer #1: No

Reviewer #2: **Yes: ** sina trighi

---

## [Editor Report · Acceptance letter]

PONE-D-24-23639R3

PLOS ONE

Dear Dr. wang,

I'm pleased to inform you that your manuscript has been deemed suitable for publication in PLOS ONE. Congratulations! Your manuscript is now being handed over to our production team.

Kind regards,

on behalf of

Dr. Zeashan Hameed Khan

Academic Editor

PLOS ONE